# Labellum Features and Chemical Composition of Floral Scent in *Bulbophyllum carunculatum* Garay, Hamer & Siegrist (Section *Lepidorhiza* Schltr., Bulbophyllinae Schltr., Orchidaceae Juss.)

**DOI:** 10.3390/plants12071568

**Published:** 2023-04-06

**Authors:** Natalia Wiśniewska, Marek Gołębiowski, Agnieszka K. Kowalkowska

**Affiliations:** 1Department of Plant Cytology and Embryology, University of Gdańsk, Wita Stwosza 59, 80-308 Gdańsk, Poland; natalia.wisniewska@ug.edu.pl; 2Laboratory of Analysis of Natural Compounds, Department of Environmental Analytics, Faculty of Chemistry, University of Gdańsk, Wita Stwosza 63, 80-952 Gdańsk, Poland; marek.golebiowski@ug.edu.pl

**Keywords:** micromorphology, ultrastructure, GC/MS, sapromyophily, fly-pollination syndrome, fragrance, attractants, pollination

## Abstract

The vast majority of fly-pollinated *Bulbophyllum* species use a combination of visual and olfactory clues to mimic food sources and brood/oviposition sites of pollinators. The aims of the present work were to characterize the floral secretory tissue and the floral scent and compare them with those previously described in *B. echinolabium*. Based on the histochemical results, the labellar secretion in *B. carunculatum* is the protein-rich mucilage. The adaxial epidermal cells of the labellum showed typical features of secretory activity. Plastids contained plastoglobuli, which are thought to be the places for scent production in osmophores. Juxtaposed with FeCl_3_ staining, the presence of dihydroxyphenolic globules in the cytoplasm of the epidermis and sub-epidermis was confirmed. Phenolic derivatives were also described with GC/MS analysis of the floral scent. The number of aromatic compounds and hydrocarbons was indicated in the floral scent of *B. carunculatum*. Moreover, pregnane-3,20-dione, occurring in the highest percentage in the floral fragrance of *B. carunculatum*, is a biologically active, 5-alpha-reduced metabolite of plasma progesterone. Progesterone is a mammalian gonadal hormone, but, like other steroid hormones, has been found in plants as intermediates in different biosynthetic pathways. The research on biosynthesis and functions of progesterone and its derivatives in flowers is still lacking.

## 1. Introduction

Insect pollinators are attracted by a combination of visual and olfactory clues. Deceptive pollination evolved in 4–6% of angiosperms ([1] and references therein) and relies on the inability of pollinators to distinguish between a true resource (e.g., mating partners, brood sites, and food) and the flower/inflorescence that imitates the reward [1,2,3]. Fly-pollinated species mimic food sources [4], brood/oviposition sites [3,5,6], or mating partners [7] using floral scents (sometimes fruity or malodorous, but they most often smell of carrion or decaying material), flower color (dull cream or yellow-green to purple-brown and frequently spotted), mobile floral parts (e.g., movable lip appendages or hair and trichomes), and glistening surface (covered in secretion) [8,9,10]. There is increasing evidence that scent plays a central role in oviposition site floral mimicry systems. Volatile organic compounds (VOCs) emitted by fly-pollinated flowers are often identical to those recorded from samples of carrion or dung [11,12,13]. The floral scent is a diverse blend of low molecular weight compounds, mostly lipophilic, emitted from flowers into the surrounding atmosphere. Alone, or with visual cues, floral scents provide chemical signals to pollinators, thereby playing crucial roles in plant reproductive and evolutionary success [14,15]. Recent studies [13] show evidence of phylogenetic constraint and plasticity of inflorescence odors. The chemical composition of the floral scent of 80 species of *Amorphophallus* (Araceae) was analyzed by headspace-thermal desorption GC/MS and compared to published molecular phylogenies of the genus. Results revealed that the pressure of pollinators has been a factor influencing the evolution of odors in *Amorphophallus*, driving both the divergence of odor types in some taxa and the convergence of odor types in others.

A prominent example of fly-pollinated deceptive plants is the genus *Bulbophyllum* (Orchidaceae). The different species are visited by a wide range of dipteran families, e.g., blow flies (Calliphoridae) pollinating *Bulbophyllum lasianthum* Lindl. (section *Beccariana* Pfitz.), *B. lobbii* Lindl. (section *Sestochilos* Benth and Hook.f.), and *B. virescens* J.J. Sm. (section *Beccariana*) [16], flesh flies (Sarcophagidae) pollinating *B. mandibulare* Rchb.f. (section *Lepidorhiza* Schltr.) [17], *B. subumbellatum* Ridl., and *B. virescens* [16], both from section *Beccariana*, or fruit flies, such as *Bactrocera* spp., reported in *B. baileyi* F. Muell. (section *Sestochilos*) [18], but often information on the actual pollinators is lacking.

*Bulbophyllum* Thouars (subtribe Bulbophyllinae Schltr.) is a large, pantropical orchid genus, containing approximately 2200 species. It occurs mainly in the Palaeotropical region of Southeast Asia, but the centers of distribution are considered Madagascar and New Guinea [19]. Section *Lepidorhiza* comprises 28 species, with Borneo, Sulawesi, the Philippines, and New Guinea as the main centers of distribution, growing in lowland or montane forests at elevations up to 1900 m a.s.l. [6,19]. The characteristic feature of *Lepidorhiza* species is the secretory tissue located superficially in the well-defined, longitudinal lip groove, which is a highly conservative lip feature among *Bulbophyllum* [20,21]. *Bulbophyllum carunculatum* Garay, Hamer, and Siegerist is found in primary and remnant forests in Sulawesi (Celebes), at elevations of 800 to 900 m. The plants are upright, sympodial epiphytes, with short rhizomes. Flowers are medium-sized, and the flower color is usually golden-yellow, with dark purple, almost black, three-lobed, fleshy labellum. Flowers of *B. carunculatum* are known to produce a mild, rather unpleasant odor.

These studies are the continuation of an anatomical survey of species from the *Lepidorhiza* section. The aims of the present work are: (a) to verify the presence of secretion in flowers, (b) to examine the secretory tissue in detail, (c) to compare the anatomical similarities and differences in the micromorphology, anatomy, and ultrastructure of secretory tissue with previously published *Bulbophyllum* species (Lepidorhiza) [6,22], and (d) to characterize the floral scent and compare it with those previously described in *B. echinolabium* [6].

## 2. Results

Flowers of *B. carunculatum* (Figure 1a–c) were around 10 cm-long, respectively. The tepals’ color was yellowish, whereas the labellum was dark-red, glistening, fleshy, around 2 cm-long, with the lateral lobes—relatively small and forwardly pointing—and attached to the base of the floral column by a small springy hinge. The lip was characterized by wart-like outgrowths and a shallow longitudinal groove at the base. Elongated, wart-like outgrowths in the basal lip part were multicellular, up to 500 µm (Figure 2a), covered by secretory residues (Figure 2b), whereas in the middle part, they were wider and shorter (Figure 2c), with the cuticle forming strands at the centers of cells (Figure 2d). Outgrowths in the apical part were less prominent (Figure 2e), also covered by secretory residues (Figure 2f).

On the transverse section, labellum (Figure 3a,b) contained the single-layer epidermis, a few layers of sub-epidermal cells, and deeply located mesophyll-like parenchyma consisting of larger, more vacuolated cells, through which ran numerous collateral vascular bundles. Numerous large idioblasts containing raphides were also noted in the sub-epidermis and upper parenchyma (Figure 3a). The sub-epidermis consisted of a few layers of cells containing dense parietal cytoplasm with organelles and enlarged nuclei (Figure 3b,c). Histochemical results obtained from the ABB test showed an abundance of proteins in the epidermis and sub-epidermis, as well as on the surface of both the adaxial and abaxial epidermis (Figure 3d). Similarly, the test with ruthenium red revealed the presence of mucilage on the surface of the adaxial and abaxial epidermis, which indicates the heterogeneous nature of the secretion (Figure 3e). The test for dihydroxyphenols detected phenolic compounds in the cytoplasm, possibly in plastids, of the abaxial epidermis (Figure 3f,g), and of cells located near the vascular bundles (Figure 3g). The thin lipid layer on cells and single-lipid drops in the epidermis and sub-epidermis were detected using the SBB test (Figure 3h). The PAS method revealed starch grains occurring in the whole labellum tissue, most profusely in the sub-epidermis and ground parenchyma, which depends on the flower stage of the anthesis (Figure 3i). The test with Auramine O failed to detect any unsaturated acidic waxes (not illustrated).

The TEM studies of the labellum showed metabolically active cells of the epidermis and sub-epidermis, containing dense cytoplasm (Figure 4a–f), enlarged nuclei (Figure 4b), numerous mitochondria (occasionally in the degenerative stage) (Figure 4a–c and Figure 5a–c), profiles of rough (RER) and smooth (SER) endoplasmic reticulum (Figure 4c,e), singular, fully developed dictyosomes (Figure 4d,e), and lipid droplets. Euchromatin was dominant in the cell nucleus, and small amounts of heterochromatin were concentrated near the nuclear envelope (Figure 4b). Additionally, an abundance of plastids with a large electron-dense body, plastoglobuli, tubules, and starch grains were present (Figure 4a,c and Figure 5b). The noteworthy feature was the presence of small vacuoles with osmiophilic material (possibly tannins) (Figure 4a–c and Figure 5b), besides autophagic vacuoles (Figure 4a), and numerous vesicles, often fusing with plasmalemma (Figure 4a and Figure 5a,c), as well as myelin-like figures occurring both in the cytoplasm of the adaxial epidermis (Figure 4d) and sub-epidermis of the labellum.

The periplasmic spaces with flocculated secretory material and numerous vesicles were present in the adaxial epidermal cells of the lip (Figure 4a and Figure 5a–d). The vesicles derived from dictyosomes in the cytoplasm and vesicles in periplasmic spaces were of similar sizes (Figure 4a and Figure 5a,d). The outer periclinal wall of the adaxial epidermis of the lateral lobe was heterogeneous, with a relatively thick cuticle with microchannels transporting secreted materials (Figure 5f). In contrast, in the adaxial (inner) epidermis of the middle lobe, discharge of secretion onto the epidermal surface probably resulted from stretching and tearing of the cuticle (Figure 5d,e), with sometimes large amounts of heterogeneous materials (Figure 5c,d), where osmiophilic content corresponded to plastoglobuli in plastids.

All these results are summarized and compared with previously published species from the *Lepidorhiza* section in Table 1.

Living flowers emitted a mild, slightly fetid scent. Of the 18 organic compounds identified in the dichloromethane labellum extract of *B. carunculatum* lip (Table 1), the pregnane-3,20-dione (30.43%), hexadecane (11.69%), and ester of bezoic (11.29%) and benzenpropanoic acids (12.89%) were the most abundant. The analysis of the volatile fraction revealed the presence of described Diptera attractants (hexadecane, tridecane, humulene).

## 3. Discussion

In terms of labellar macromorphology and anatomy, the results here were almost identical to those of the other representatives of the *Lepidorhiza* section [6,22], as well as other *Bulbophyllum* species, e.g., [10,20,40,41,42], which indicates the previously described highly conservative character of the *Bulbophyllum* labellum. Therefore, a well-balanced trilobed labellum of *B. carunculatum* was attached to the column foot by a springy hinge. The labellum was glistering and covered with warts. The shallow and poorly defined longitudinal groove closely resembled the adaxial labellar groove of the neotropical representatives of *Bulbophyllum* [20,21] and three West African species [40], thereby fitting into the second of the third distinguished types of labellar organization, where the groove is ill-defined or absent, and those species are likely to be myophilous [40]. *Bulbophyllum* species vary in their labellar secretion, from nectar in *B. echinolabium* [6] and *B. schinzianum* [41], to protein-rich mucilage in *B. weberi* [42] and four species of Asian *Bulbophyllum* sect. *Racemosae* [10,41], and lipid-rich labellar secretions in *B. lupulinum* [40] and *B. levanae* [22]. In *B. carunculatum*, large amounts of heterogeneous residues of secreted material were present on the surface of the adaxial labellar epidermis. Testing with ruthenium red revealed the presence of mucilage, whereas ABB staining showed a great number of proteins in the adaxial and abaxial labellar epidermis, as well as on its surface; therefore, we assume that the labellar secretion in *B. carunculatum* is the protein-rich mucilage. The presence of proteins in secreted material was previously noted in a number of fly-pollinated species, which is regarded to be luring protentional pollinators. Observations of Sarcophagidae, whose representatives are described as pollinators in Apocynaceae, Iridaceae, and Orchidaceae [10,12,17], show that flies with protein deprived or without access to protein or amino acid sources chose an amino acid-containing nectar over sucrose-only nectar [43]. Furthermore, proteins are required for egg maturation [44] and larvae of carrion flies do not develop in herbivorous feces, probably due to the low/absent protein content [3,45], which is also reflected in the absence of sulfur-containing compounds in the volatile emission.

In *B. carunculatum*, the adaxial epidermal cells of the labellum showed typical features of secretory activity. Dense cytoplasm contained enlarged nuclei, numerous mitochondria associated with high metabolic cell activity, profiles of endoplasmic reticulum, and fully developed dictyosomes, together with numerous vesicles, often fusing with plasmalemma. Plastids contained plastoglobuli, which are thought to be the places for scent production in osmophores [42,46]. Juxtaposed with FeCl_3_ staining, the presence of dihydroxyphenolic globules in the cytoplasm of the epidermis and sub-epidermis was confirmed. Phenolic derivatives were also described with GC/MS analysis of the floral scent. Floral volatile organic compounds (VOCs) are synthesized in plastids’ plastoglobuli, transported to the intra0plastidal membranes, crossing the plastid envelope to the profiles of ER (or migrating independently in the cytoplasm), and finally, transported to plasmalemma, where they secreted on the surface of cells [6,41,42,47,48]. The periplasmic spaces, where the secreted material gathers, were present in the adaxial epidermal cells of lips, between the cell wall and the protoplast. The formation of periplasmic spaces is probably associated with a granulocrine type of secretion. In the conventional approach, the synthesized substances are transported via vesicles outside the protoplast, where they gather in the periplasmic spaces underneath the cell wall and then, as a result of a pressure increase in the protoplast, are secreted outside [49]. Subcuticular spaces were noted in a few other species: in the labellar epidermis of three *Bulbophyllum* species [6,22] and *Epipactis helleborine* [50], the elaiophores in *Oncidium trulliferum* flowers [51], and in the adaxial epidermal cells of the corona lobes of *Stapelia scitula* (Apocynaceae) [52]. In *B. carunculatum*, the exudation is transported outside the cell through micro-channels in the cuticle in the lateral lobe or secreted material traverses the cell wall and is released by cuticle rupture in the middle lobe. The combination of periplasmic space and microchannels was previously described in another species from the *Lepidorhiza* section, *B. levanae* [22].

As mentioned above, in the conventional approach, secretion is transported via vesicles and may be ER-derived and sorted by the trans-Golgi network into a plasma membrane. This pathway is highly conservative among organisms. However, recent studies on the plant secretory pathway report another way: unconventional protein secretion (UPS) [53]. In UPS, secreted protein either bypasses the conventional endomembrane compartments (ER, trans-Golgi network) and is directed to multivesicular bodies (MVBs) before insertion into the plasma membrane or, in other pathways, through to the specialized exocyst-positive organelle (EXPOs), a double membrane-bound organelle identified by the exocyst protein EXO70E2 [54]. Histochemical studies of *B. carunculatum* showed an abundance of proteins in the epidermal and sub-epidermal cells of the labellum as well as on the adaxial surface of the epidermis. Ultrastructural analysis did not confirm the presence of MVBs in the epidermal cells of *B. carunculatum*, unlike in other *Bulbophyllum* species [6,22]. This indicates different routes of protein transport in closely related species. However, since immunolocalization for the presence of EXPO proteins has not yet been performed, we cannot completely exclude the use of an unconventional protein secretion route in *B. carunculatum* secretory cells. However, further research is needed, focusing on the secretory pathway in epidermal cells.

The floral scent is a diverse blend of low molecular weight compounds, mostly lipophilic, emitted from flowers into the surrounding atmosphere. Alone, or with visual cues, floral scents provide chemical signals to pollinators, thereby playing crucial roles in plant reproductive and evolutionary success [14,15]. There is increasing evidence that plants use different blends of fragrance components to attract saprophagous, coprophagous, and necrophagous insects, respectively [5,12,55]. An extensive analysis of the chemical composition of the floral scent in 172 species of 36 plant families conducted by Jürgens et al. [5] revealed that species mimicking oviposition sites can be divided into three chemical groups: (1) Species dominated by dimethyl disulfide and dimethyl trisulfide, compounds that are normally products of microbial degradation of the amino acids methionine and cysteine. (2) Species dominated by aromatic compounds and hydrocarbon aldehydes, where these compounds are typically found as products of microbial degradation of lipids and amino acids. (3) Species dominated by hydrocarbon acids, alcohols, and esters, where these compounds are often found as fermentation products of carbohydrates and amino acids ([56] and references therein). The number of aromatic compounds (e.g., methyl 3-hydroxybenzoate, ethyl 4-ethoxybenzoate) and hydrocarbons (e.g., hexadecane, tetradecane, dodecane, 3-methyl-2-nonene, 3-ethyl-2-methyl-1-heptene) indicates that the floral scent of *B. carunculatum* belongs to the second chemical group. As suggested by Ollerton and Raguso [57] (see also [5]), particular blends of volatiles constituted by ‘adaptive peaks’ in the ‘floral phenotype space’ play a pivotal role in floral visitations of different groups of insects (saprophagous, coprophagous, or necrophagous), as they are associated with their oviposition sites. Although the genus *Bulbophyllum* is regarded to be pollinated by Diptera and Hymenoptera [9,10,16], detailed studies of pollinators have been conducted only for a few species and the results show the complexity of the flower–pollinator relationship. Blowflies (Calliphoridae) pollinate both malodorous flowers of *Bulbophyllum lasianthum* and *B. virescens* (section *Beccariana*), together with *B. lobbii* (section *Sestochilos*), which produce a fragrance reminiscent of over-ripe fruit [16]. However, another species from section *Sestochilos*, *B. baileyi*, is pollinated by fruit flies, such as *Bactrocera* spp. [18]. Some members of the genus are pollinated equally by both male and female dipterans, such as blowflies and flesh flies in *B. virescens* [16], while others produce a phenylpropanoid-rich, fruity, or spicy floral scent that attracts only male fruit fly pollinators [8,9,15,58,59]. Thus, it is not surprising that, although closely related, *B. carunculatum* and the previously examined *B. echinolabium* [6] have only six chemical compounds of floral scent in common and two others which are close derivatives (1-methyl-2-propyl-cyclohexane, 1-methyl-3-propyl-cyclohexane, benzoic acid, 3-ethyloxy-, ethyl ester, and benzoic acid, and 4-ethoxy-, ethyl ester). Differences could be a result of various habitats or different pollinators, which are not described in these species. Recent studies [13] show evidence of the phylogenetic constraint and plasticity of inflorescence odors. The chemical composition of the floral scent of 80 species of *Amorphophallus* (Araceae) was analyzed by headspace-thermal desorption GC/MS and compared to published molecular phylogenies of the genus. Results revealed that the pressure of pollinators has been a factor influencing the evolution of odors in *Amorphophallus*, driving both the divergence of odor types in some taxa and the convergence of odor types in others. Phylogenetic mapping of odors also indicates that the evolution of some odor types is likely to have been influenced by ecological factors. For example, species producing fishy odors dominated by trimethylamine and occurring in North and Northeast Borneo are not all closely related. Conversely, two sister species, *A. mossambicensis*, and *A. abyssinicus*, which are morphologically very similar and have an overlapping geographical distribution, produce odors that are very chemically different.

What also combines the chemical composition of the floral scent of the previously mentioned *Bulbophyllum* species is the presence of pregnane-3,20-dione, a metabolite of plasma progesterone in *B. carunculatum* and cholest-5-en-3-ol in *B. echinolabium* [6]. Cholest-5-en-3-ol is a derivative of cholesterol, which, in turn, is a precursor to numerous steroids, such as steroid hormones, e.g., progesterone. Pregnane-3,20-dione, occurring in the highest percentage in the floral fragrance of *B. carunculatum*, is a biologically active 5-alpha-reduced metabolite of plasma progesterone. Progesterone is a mammalian gonadal hormone, but other steroid hormones have been found in plants as intermediates in different biosynthetic pathways [60]. In *Digitalis* representatives, progesterone is one of the intermediate compounds in the biosynthetic pathway of cardiac glycosides ([61] and references therein), and enzymes responsible for the conversion of progesterone and other pregnane derivatives have also been found. Progesterone is an intermediate in cardenolide pathways and is present in plants producing cardenolides such as *Digitalis purpurea* L. and *Cheiranthus cheiri* L. [61]. In the cardenolide pathway, ∆5-3β-hydroxysteroid dehydrogenase/∆5-∆4-ketosteroid isomerase (∆5-3β-HSD) transforms pregnenolone into progesterone. Then, progesterone-5β-reductase transforms progesterone to 5β-pregnane-3,20-dione or progesterone-5α-reductase transforms progesterone to 5α-pregnane-3,20-dione. Regardless of the multiple studies, the understanding of the biological importance, biosynthesis, and functions of progesterone in plants is still relatively poor [61,62]. Most of the studies have been conducted by exogenously applying progesterone to various plant systems. Results suggest that it has a certain regulatory activity in plant growth and development, affecting both vegetative and reproductive development. Studies conducted by Iino et al. [62] allowed to identify and quantify progesterone in a range of higher plants by using GC/MS and examined its effects on the vegetative growth of plants, such as the shoot and root growth of Arabidopsis thaliana, influenced by the progesterone in a dose-dependent manner. However, research on biosynthesis and the functions of progesterone and its derivatives in flowers is still lacking.

## 4. Materials and Methods

Tissue samples were collected from fresh flowers at anthesis (first day of the opening of the flower) from the greenhouse at the University of Gdansk, Faculty of Biology (voucher number: 030722). To fix plant material, 2.5% (*v*/*v*) glutaraldehyde (GA) in 0.05 M cacodylate buffer (pH = 7.0) was used. Material for light microscopy (LM) following fixation was rinsed with cacodylate buffer, dehydrated in an ethanol series, and embedded in methylmethacrylate-based resin (Technovit 7100, Kulzer GmbH, Kulzer Technik, Wehrheim, Germany). Sections were cut with glass knives (1–3 μm-thick) using Sorvall MT 2B and a Leica EM UC 7 ultramicrotomes and mounted on glass slides. Transverse sections were presented from different portions (apical, middle, and basal of the labellum). For histochemical analysis, semi-thin control sections for light microscopy were stained with 0.05% (*w*/*v*) aqueous Toluidine Blue O (TBO, C.I. 52040) [63,64]. Aniline Blue Black (ABB, C.I. 20470) was used for the detection of water-insoluble proteins, and the Periodic Acid–Schiff reaction (PAS) to identify the presence of water-insoluble polysaccharides [65]. A 0.05% (*w*/*v*) aqueous Ruthenium Red (C.I. 77800) solution and a 10% (*w*/*v*) aqueous solution of FeCl_3_ were used to test for pectic acids/mucilage [66] and catechol-type dihydroxyphenols [67], respectively. The preparations were examined and photographed with a Nikon Eclipse E 800 light microscope and a Nikon DS-5 Mc camera using Lucia Image software (University of Gdańsk, Gdańsk, Poland). The sections, following FeCl_3_ staining, were observed using the differential interference contrast (DIC) imaging. Auramine O (C.I. 41000) 0.01% (*w*/*v*) solution in 0.05 M Tris/HCl buffer, pH = 7.2 was used to present cuticle [68], especially cutin precursors and unsaturated acidic waxes [67]. Nucleus structure was examined in sections stained with the fluorochrome 4′,6-diamidino-2-phenylindole (DAPI). Staining reactions with Auramine O and DAPI were examined with a Nikon Eclipse E800 fluorescence microscope equipped with filter B-2A (EX 450–490 nm, DM 505 nm, BA 520 nm).

After dehydration in an ethanol series, the samples were subjected to critical-point drying using liquid CO_2_, coated with gold and examined for micromorphology in scanning electron microscopy (SEM) using a Philips XL-30 at an accelerating voltage of 15–20 kV (Laboratory of Electron Microscopy, University of Gdańsk, Poland).

For studies in transmission electron microscopy (TEM), lips were fixed in, aldehyde (GA) in 0.05 M cacodylate buffer (pH 7.0). The material was then post-fixed overnight in 1% (*w*/*v*) OsO4 in cacodylate buffer in a refrigerator and finally rinsed with buffer. After 1 h in a 1% (*w*/*v*) aqueous solution of uranyl acetate, the material was dehydrated with acetone and embedded in Spurr’s resin [69]. Semi-thin sections (0.8–1 μm-thick) were mounted on glass slides and treated with a 0.3% (*w*/*v*) ethanolic solution of Sudan Black B (SBB, C.I. 26150) for lipid localization [70], observed with a Nikon Eclipse E800 light microscope. Ultrathin sections were cut on a Leica EM UC7 ultramicrotome with a diamond knife and stained with uranyl acetate and lead citrate [71]. The sections were examined in a Tecnai G2 Spirit BioTwin FEI transmission electron microscope (Laboratory of Electron Microscopy, University of Gdańsk, Poland) at an accelerating voltage of 120 kV. For scanning electron microscopy, following dehydration in an ethanol series, the flower samples were subjected to critical-point drying using liquid CO_2_ and sputter-coated with gold. They were then examined using a Philips XL-30 scanning electron microscope (SEM), at an accelerating voltage of 15–20 kV (Laboratory of Electron Microscopy, University of Gdańsk, Poland). Plant samples were extracted by immersing in 20 mL of dichloromethane for 15 min. Solvents were evaporated under nitrogen to 4 mL at room temperature and extracts were kept at 4 °C until analysis. All extracts were subjected to a derivatization process. The dried samples were silylated with 100 μL of a mixture of 99% bis(trimethylsilyl)acetamide and 1% chlorotrimethylsilane (BSTFA) for 1 h at 100 °C. The GC/MS analysis was conducted using a GC/MS QP-2010 SE (Shimadzu, Kyoto, Japan) equipped with a ZB-5 capillary column (30 m × 0.25 mm × 0.25 μm film thickness) (Zebron, Phenomenex, USA). The temperature program was as follows: 40 °C to 310 °C at 4 °C/min, and finally 10 min at 310 °C. A total of 1 μL of each extract was injected. The transfer line and injector temperatures were held at 310 °C. The ion source temperature was maintained at 200 °C. The split ratio was 1:20. The mass spectrometer was used in the electron impact (EI) mode (70 eV) and set to scan the 40–650 Da mass range. Compounds were identified as native compounds or trimethylsilyl (TMS) derivatives of these compounds. All compounds were identified on the basis of the characteristic ions and the retention time. The content of the compounds was calculated from the peak areas from the total ion current (TIC).

## Figures and Tables

**Figure 1 plants-12-01568-f001:**
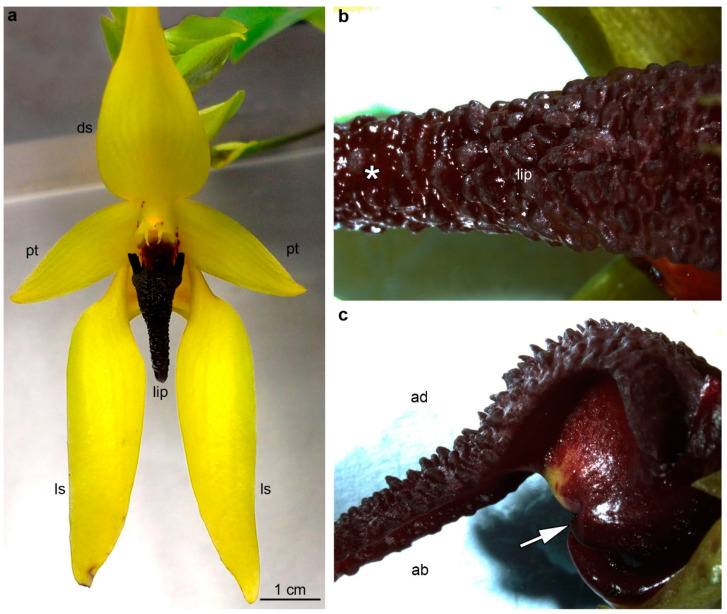
General floral morphology of *Bulbophyllum carunculatum*. (**a**) Inflorescence. (**b**) Adaxial surface of labellum with glistering secretion (asterisk). (**c**) Lateral view of the lip, the white arrow points to the hinge. ab: Abaxial (outer) surface, ad: adaxial (inner) surface, ds: dorsal sepal, ls: lateral sepal, pt: petal.

**Figure 2 plants-12-01568-f002:**
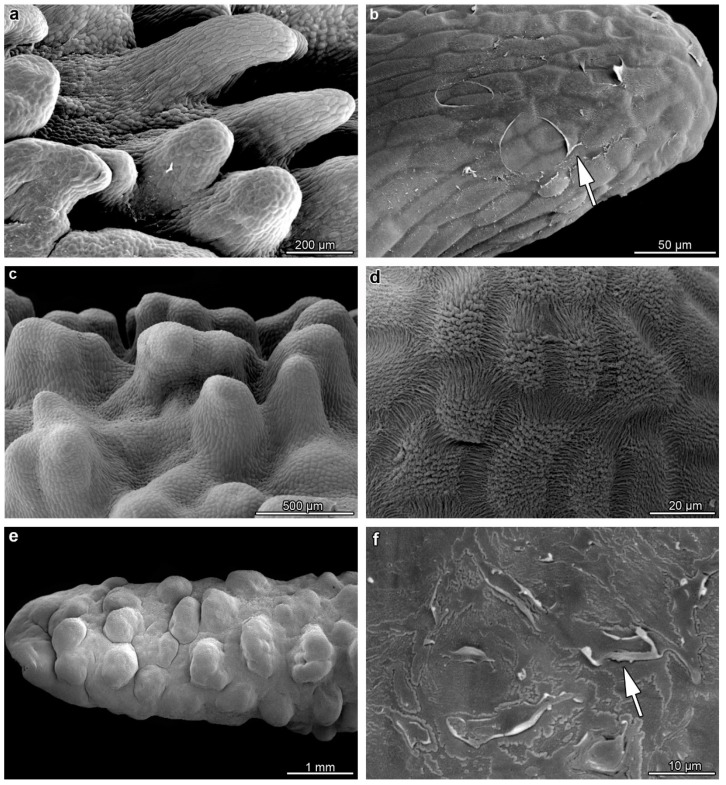
Micromorphological features of the adaxial surface of the labellum of *B. carunculatum*. (**a**,**b**) Elongated, multicellular outgrowths covered with ruptured cuticle ((**b**), white arrow) in the basal part of the lip. (**c**,**d**) Middle part of the lip with shorter, wide outgrowths with a strongly striated cuticle (**d**). (**e**,**f**) Apical part of the lip with secretory residues (white arrow) noted on the wart-like outgrowths.

**Figure 3 plants-12-01568-f003:**
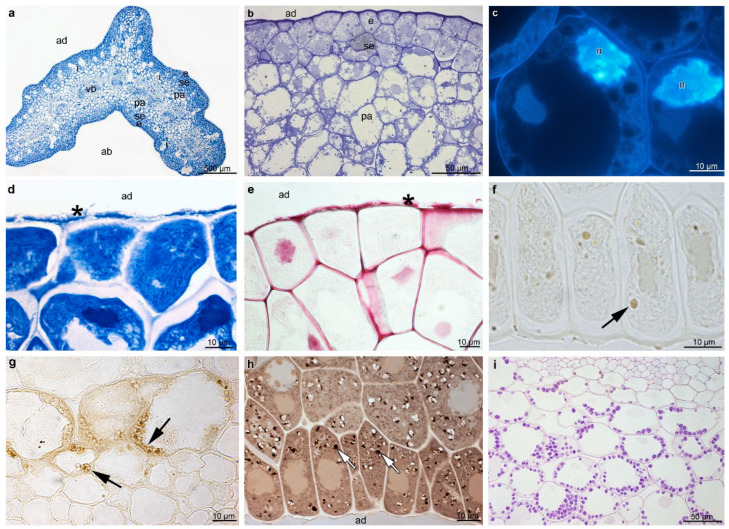
Histochemical features of the B. carunculatum (transverse sections). (**a**) General view (stained with ABB) showing a single layer of the epidermis, a few layers of sub-epidermal cells with idioblasts, and several collateral vascular bundles in the ground parenchyma. (**b**) Detail of the adaxial surface of the middle lobe, with dense cytoplasm and enlarged nuclei in the epidermis and sub-epidermis (TBO). (**c**) Enlarged nuclei in the adaxial sub-epidermis (DAPI). (**d**) The epidermis and sub-epidermal tissue stained intensively for proteins (ABB), note the secretory material on the surface (black asterisk). (**e**) I Staining with ruthenium red revealed the presence of mucilage (black asterisk) on the adaxial epidermis of the middle lobe. (**f**,**g**) Dihydroxyphenols inclusions (FeCl_3_) in the cytoplasm (possibly plastids) of epidermal cells (**f**) and in singular cells near the vascular bundle (**g**) (black arrows). (**h**) Lipid droplets (white arrows) in the adaxial epidermis and sub-epidermis. (**i**) Abundance of starch grains in the ground parenchyma (PAS). ab: Abaxial surface, ad: adaxial surface, e: epidermis, i: idioblast, n: nuclei, pa: parenchyma, se: sub-epidermis, vb: vascular bundle.

**Figure 4 plants-12-01568-f004:**
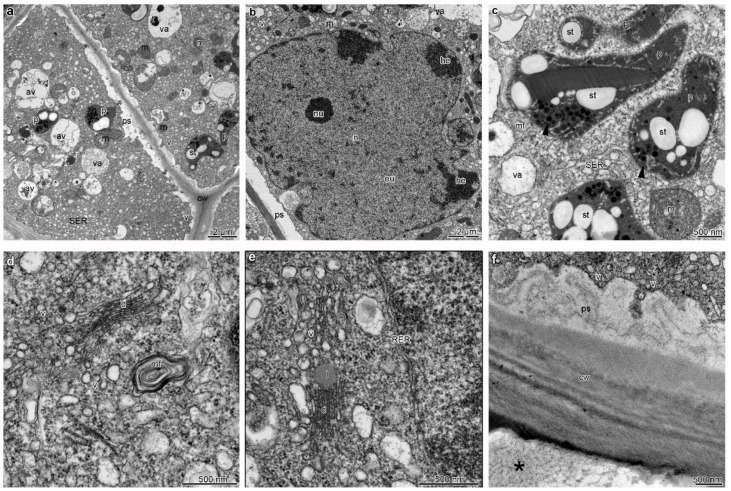
Ultrastructural details (TEM) of the middle lobe of the labellum of B. carunculatum, showing secretory epidermal cells with dense cytoplasm (**a**–**e**), enlarged nuclei (**b**), numerous small vacuoles, occasionally with osmiophilic (tannin-like) material (**a**,**c**), an abundance of mitochondria (**a**–**c**), plastids with starch grains and plastoglobuli (**a**,**d**), fully developed dictyosomes (**d**,**e**), profiles of smooth (SER) (**c**) and rough endoplasmic reticulum (RER, (**e**)), lipid droplets (**e**), and periplasmic space with flocculated secretory material, numerous vesicles building into plasmalemma, (**f**) and residues of secretion on the epidermis surface (black asterisk). av: Autophagic vacuole, cw: cell wall, d: dictyosome, eu: euchromatin, he: heterochromatin, l: lipid droplet, m: mitochondrion, ml: myelin-like figures, n: nuclei, nu: nucleolus, p: plastid, ps: periplasmic space, va: vacuole, st: starch grain, t: tannin-like material, v: vesicle.

**Figure 5 plants-12-01568-f005:**
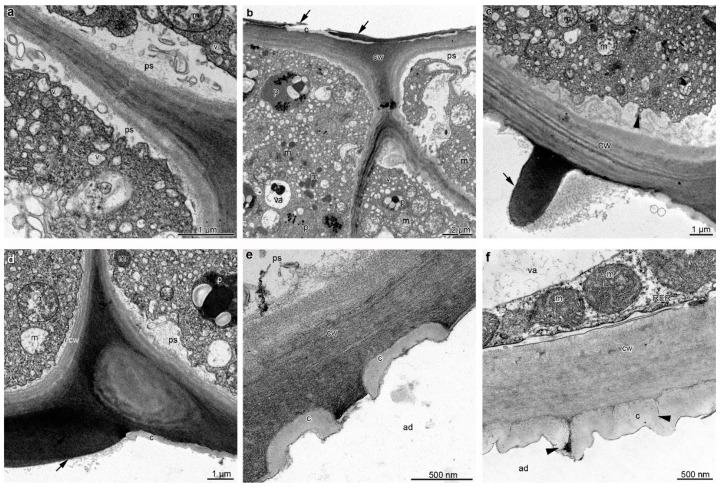
Ultrastructure of the adaxial epidermis of the middle (**a**–**e**) and lateral (**f**) lobes of the labellum of B. carunculatum (TEM). (**a**–**c**) Cells with periplasmic space, flocculated secretory material, and numerous vesicles building into plasmalemma ((**c**), black arrowhead). (**b**–**e**) Adaxial epidermis with ruptured cuticle and residues of secretion on the surface (**b**–**d**) (black arrowheads). (**f**) Cell wall with a cuticle with microchannels with secretion (black arrowheads). c: Cuticle, cw: cell wall, m: mitochondrion, p: plastid, ps: periplasmic space, va: vacuole, v: vesicle.

**Table 1 plants-12-01568-t001:** Chemical composition of labellum of *Bulbophyllum carunculatum* (GC/MS analysis, extract from dichloromethane) in the context of properties and occurrence in other species.

	TR: Retention Time	Relative Content (%)	*Bulbophyllum carunculatum* Labellum	
1	3.900	0.50	3,5,5-trimethyl-1-hexene,	Alkane (hydrocarbons)
2	4.670	0.46	2,3-dimethyl-2-heptene (trimethyl-hexene)	Alkane (hydrocarbons)
3	8.071	3.94	3-ethyl-2-methyl-1-heptene	Alkane (hydrocarbons) Noted in the floral scent of *B. echinolabium* [6] and in the scent of ox carcass [23]
4	8.179	0.74	1,2,3,4,5-pentamethyl-cyclopentane (pentamethyl-cyclopentane)	Alkane (hydrocarbons) Noted in *Maxillaria sanguinea* (Orchidaceae) [24]
5	9.504	4.50	3-methyl-2-nonene,	Alkene (hydrocarbons) Noted in an aroma of cured pork [25]
6	9.614	1.06	1-methyl-2-propyl-cyclohexane,	Alkane (hydrocarbons) Noted in the floral scent of Orchidaceae: *B. echinolabium* [6] and *M. sanguinea* [24]
7	21.098	1.10	Dodecane	Aliphatic alkane (hydrocarbons) Noted in the floral scent of Orchidaceae: *B. echinolabium* [6], *M. sanguinea* [24], and Aristolochiaceae: *A. gigantea* [26] Pheromone and allomone Hymenoptera, Coleoptera, found in *Araceae* and *Orchidaceae* [27]
8	24.891	5.14	Tridecane	Aliphatic alkane (hydrocarbons) Noted in the floral scent of Orchidaceae: *B. echinolabium* [6], *M. sanguinea* [24], Aristolochiaceae: *A. gigantea* [26], and Apocynaceae: *Huernia hystrix* [12] Attractant of Diptera (Chloropidae and Milichiidae), pheromone and allomone of Hymenoptera, Heteroptera, and Coleoptera [27]. Found mainly in Araceae, Arecaceae, Orchidaceae (*Coryanthes* spp., *Cymbidium* spp., *Ophrys* spp., *Phalaenopsis* spp.), Magnoliaceae, and Moraceae [27]
9	28.404	3.27	Tetradecane	Aliphatic alkane (hydrocarbons) Noted in the floral scent of Orchidaceae: *B. echinolabium* [6], *M. vulganica* [24] Pheromone and allomone Hymenoptera, Heteroptera, Coleoptera, and Astigmata [27]
10	30.146	0.20	Humulene	Sesquiterpenoid (hydrocarbons) Noted in the floral scent of fly-pollinated Araceae: *A. maculatum* and *A. italicum* ssp. *neglectum* [28], *Sauromatum guttatum* [29], and Orchidaceae: *B. variegatum* [30]. Noted also in small amounts in horse dung [12] pheromone Diptera, Hymenoptera, and Coleoptera, and attractant Hymenoptera, Heteroptera, and Coleoptera [27]. Humulene gives a woody, earthy, and spicy fragrance
11	30.621	2.83	2,5-di-tert-butyl-1,4-benzoquinone (DTBBQ)	Dicarbonyl (diketones), a 2,5-disubstituted quinone. Noted in the floral scent of fly-pollinated *Carraluma europea* [31] It is an antibacterial compound, neurotoxic for humans [32], with antimicrobial activity against *Bacillus cereus* [33]. Quinones are highly electrophilic compounds that are dietary plant components and arise also from the metabolism of benzene, phenols, and other aromatics, including polycyclic aromatics of environmental origin [34]
12	30.905	4.92	Benzoic acid, 3-hydroxy-, methyl ester (Methyl 3-hydroxybenzoate)	Benzoate (an aromatic carboxylic acid ester). Esters of benzoic acids have a pleasant, intense fragrance and are used in the perfume industry [35]
13	32.250	3.89	Phenol, 3,5-di-tert-butyl-	Phenol derivative
14	32.538	11.29	Benzoic acid, 4-ethoxy-, ethyl ester (Ethyl 4-ethoxybenzoate)	Benzoate (an aromatic carboxylic acid ester). Esters of benzoic acids have a pleasant, intense fragrance and are used in the perfume industry Noted in male wing androconia of the neotropical butterfly *Heliconius* [36]
15	34.796	11.69	Hexadecane	Hydrocarbons (aliphatic alkane). Noted in the floral odor of *Arum maculatum* [28] and *Sauromatum guttatum* [29] Odor characteristic: fusel-like, fruity, sweet attractant of papaya fruit fly (*Toxotrypana curvicauda*), the pheromone of Hymenoptera, Lepidoptera, Diptera, Homoptera, Coleoptera, Iguanidae, and Rodentia. Found in many Orchidaceae (i.e., *Aerangis* spp., *Cattleya* spp., *Dendrobium* spp., *Odontoglossum* spp., *Orphys* spp.), Cactaceae, and Araceae [27]
16	37.090	1.15	n-hexyl salicylate (Hexyl 2-hydroxybenzoate)	Benzoic acid ester (derivative of benzoic acid) Used as flavoring and odor agents. Scent described as fresh, sweet, herbal, pleasant, natural, and floral [37]
17	44.293	12.89	Benzenepropanoic acid, 3,5-bis(1,1-dimethylethyl)-4-hydroxy, methyl ester (Methyl 3-(3,5-di-tert-butyl-4-hydroxyphenyl)propionate)	Phenol derivative Noted in the floral scent of Orchidaceae: *B. echinolabium* [6]
18	61.331	30.43	Pregnane-3,20-dione (5-alpha-pregnan-3,20-dione)	Lipids, steroid compounds. 5-alpha-pregnan-3, 20-dione is part of the protein modification, steroid hormone biosynthesis, and prostate cancer pathways. It is a substrate for probable polyprenol reductase, 3-oxo-5-alpha-steroid 4-dehydrogenase 1, and 3-oxo-5-alpha-steroid 4-dehydrogenase 2 [38]. A biologically active 5-alpha-reduced metabolite of plasma progesterone. It is the immediate precursor of 5-alpha-pregnan-3-alpha-ol-20-one (allopregnanolone), a neuroactive steroid that binds with the gaba(a) receptor. A pregnane was found in the urine of pregnant women and sows. It has anesthetic, hypnotic, and sedative properties [39].

## Data Availability

Data are available upon request from the corresponding author.

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
