# Peer review of "Labellum Features and Chemical Composition of Floral Scent in Bulbophyllum carunculatum Garay, Hamer & Siegrist (Section Lepidorhiza Schltr., Bulbophyllinae Schltr., Orchidaceae Juss.)"

_plants, 2023, doi:10.3390/plants12071568_

Round 1

Reviewer 1 Report

This manuscript describes characterization of the floral secretory tissue and the floral scent of Bulbophyllum sp.

The experiments herein seem to be properly done with adequate experts. The paper is totally well-written and the results would be much of interest to the readers.

My evaluation is that this paper would be appropriate for Plants just as it is.

Author Response

Thank you for your time in reviewing the manuscript and positive decision.

With regards,

Agnieszka Kowalkowska

Reviewer 2 Report

The manuscript is very interesting and well discussed. English is flawless.

I suggest very few corrections:

- d) to characterize the floral scent and compare it with previously described in B. echinolabium. Please, add the proper reference at the end of the sentence; 

- Figure 1a,b,c: Bar scale is missing

- FIgure 3c: "c" is missing at the top left of the photo. I suggest to use a white font; 

- Figure 4: letters and bar scales are barely visible. I suggest to use a white font; 

- Living flowers emitted a mild, slightly fetid scent. Of the 18 organic compounds identified in the dichloromethane labellum extract of B. carunculatum lip (Table 1). This is not Table 1: please rename it (in the text and in the supplementary materials) as Table S1, since it is the first table (1) included in supplementary meterials (S). Actually, I suggest to include the Chemical composition of labellum of Bulbophyllum carunculatum in the main text and to move  "A comparison of macromorphology, anatomy, micromorphology, and ultrastructure of labellum of B. levanae, B. nymphopolitanum [22], B. echinolabium [6], and B. carunculatum" (Table 1) in the supplementary materials. 

- first line of page 6: B. carunculatum require italics; 

- Some members of the genus are pollinated equally by both male and female dipterans, such as blowflies and flesh flies in B. virescens. B. virescens should be in italics

Author Response

Thank you for your time in reviewing our work. Our replies are placed below.

Reviewer 2:

- d) to characterize the floral scent and compare it with previously described in B. echinolabium. Please, add the proper reference at the end of the sentence; - added

- Figure 1a,b,c: Bar scale is missing. - The scale bar is added to Fig. 1a. We don't have the scales to photos (Fig. 1b-c).

- FIgure 3c: "c" is missing at the top left of the photo. I suggest to use a white font; - added

- Figure 4: letters and bar scales are barely visible. I suggest to use a white font; - the visibility of letters and scales is improved

- Living flowers emitted a mild, slightly fetid scent. Of the 18 organic compounds identified in the dichloromethane labellum extract of B. carunculatum lip (Table 1). This is not Table 1: please rename it (in the text and in the supplementary materials) as Table S1, since it is the first table (1) included in supplementary meterials (S). Actually, I suggest to include the Chemical composition of labellum of Bulbophyllum carunculatum in the main text and to move  "A comparison of macromorphology, anatomy, micromorphology, and ultrastructure of labellum of B. levanae, B. nymphopolitanum [22], B. echinolabium [6], and B. carunculatum" (Table 1) in the supplementary materials. - corrected. Table 1 is now a supplementary material Table S1; Table 2 is now Table 1, placed in the text.

- first line of page 6: B. carunculatum require italics; - corrected

- Some members of the genus are pollinated equally by both male and female dipterans, such as blowflies and flesh flies in B. virescens. B. virescens should be in italics - corrected